# Circulation of Bluetongue Virus Serotypes 1, 4, 8, 10 and 16 and Epizootic Hemorrhagic Disease Virus in the Sultanate of Oman in 2020–2021

**DOI:** 10.3390/v15061259

**Published:** 2023-05-27

**Authors:** Emmanuel Bréard, Lydie Postic, Mathilde Gondard, Cindy Bernelin-Cottet, Aurélie Le Roux, Mathilde Turpaud, Pierrick Lucas, Yannick Blanchard, Damien Vitour, Labib Bakkali-Kassimi, Stéphan Zientara, Wafaa Al Rawahi, Corinne Sailleau

**Affiliations:** 1UMR 1161 VIROLOGIE, INRA, ENVA, ANSES, Laboratory for Animal Health, Paris Est University, 94701 Maisons-Alfort, France; lydie.postic@anses.fr (L.P.); mathilde.gondard@anses.fr (M.G.); cindy.bernelin-cottet@anses.fr (C.B.-C.); mathilde.turpaud@anses.fr (M.T.); damien.vitour@anses.fr (D.V.); labib.bakkali-kassimi@anses.fr (L.B.-K.); stephan.zientara@anses.fr (S.Z.); corinne.sailleau@anses.fr (C.S.); 2Laboratory of Ploufragan, ANSES, Unit of Viral Genetics and Biosafety, 22440 Ploufragan, France; aurelie.leroux@anses.fr (A.L.R.); pierrick.lucas@anses.fr (P.L.); yannick.blanchard@anses.fr (Y.B.); 3Department of Biology, College of Science, Sultan Qaboos University, Muscat P.C. 123, Oman; s69556@student.squ.edu.om; 4Central Laboratory of Animal Health, Ministry of Agriculture, Fisheries Wealth and Water Resources, Muscat P.C. 100, Oman

**Keywords:** bluetongue, epizootic hemorrhagic disease, the Sultanate of Oman, BTV serotype 8

## Abstract

The circulation of Bluetongue (BT) and Epizootic Hemorrhagic Disease (EHD) in the Middle East has already been reported following serological analyses carried out since the 1980s, mostly on wild ruminants. Thus, an EHD virus (EHDV) strain was isolated in Bahrain in 1983 (serotype 6), and more recently, BT virus (BTV) serotypes 1, 4, 8 and 16 have been isolated in Oman. To our knowledge, no genomic sequence of these different BTV strains have been published. These same BTV or EHDV serotypes have circulated and, for some of them, are still circulating in the Mediterranean basin and/or in Europe. In this study, we used samples from domestic ruminant herds collected in Oman in 2020 and 2021 for suspected foot-and-mouth disease (FMD) to investigate the presence of BTV and EHDV in these herds. Sera and whole blood from goats, sheep and cattle were tested for the presence of viral genomes (by PCR) and antibodies (by ELISA). We were able to confirm the presence of 5 BTV serotypes (1, 4, 8, 10 and 16) and the circulation of EHDV in this territory in 2020 and 2021. The isolation of a BTV-8 strain allowed us to sequence its entire genome and to compare it with another BTV-8 strain isolated in Mayotte and with homologous BTV sequences available on GenBank.

## 1. Introduction

Bluetongue (BT) and Epizootic Hemorrhagic Disease (EHD) are vector-borne diseases affecting domestic and wild ruminants caused by two orbiviruses which can induce significant production losses in livestock industries [1,2]. BT virus (BTV) and EHD virus (EHDV) are arboviruses mainly transmitted by hematophagous insects (*Culicoides* midges) [3]. It is assumed that the epidemiological characteristics of EHDV reflect those of BTV. Before the 2000s, BTV and EHDV were restricted to between 40° and 50° North latitude and 35° South latitude, these latter being confined to the tropical and subtropical regions [1,2]. Since the beginning of the 21st century, the situation has changed, especially in Europe where BTV has been detected up to 60° north latitude [4].

BTV has a genome of 10 double-stranded RNA segments (S) encoding 7 structural (viral protein 1 (VP1) to VP7) and 6 non-structural proteins (NS1-NS4, NS3a and NS5) [5,6]. BTV serotypes are recognized on the basis of specific interactions between neutralizing antibodies and two structural proteins (VP2 and VP5) forming an outer-capsid layer.

The occurrence of BTV in the Middle East region is already known [7,8,9]. In Saudi Arabia and the United Arab Emirates [9], it has been shown that these viruses were enzootic following serological analyses carried out mostly on wild ruminants. In the Sultanate of Oman, ELISA and serum neutralization test identified at least three BTV serotypes (BTV-3, 4 and 22) present in 1987–1988 [7,8]. In 2009, BTV-1, 4, 8 and 16 were isolated in Oman (www.reoviridae.org/dsRNA_virus_proteins/outbreaks.htm#AHS-2010/, accessed on 14 May 2023), and in 2010, BTV-1 and 16 were detected in four Oryx antelopes imported from Oman to Croatia [10]. To our knowledge, no genomic sequence of these different BTV strains have been published to date.

In 1983, an EHDV was isolated in Bahrain. This virus was first classified as serotype 3, and since 2009, it is recognized that this strain is serotype 6 [11]. In Israel, several serotypes of this virus have been detected (serotypes 1, 6 and 7). In the Maghreb countries, the same serotypes have been detected with, since 2021 (in Tunisia), serotype 8 [12]. This strain was detected in Spain, Sardinia and Sicily in 2022, ending the free status of Europe with respect to this disease [13].

In Europe and in the Mediterranean Basin, continuous BTV incursions are observed involving different serotypes: BTV-1, 2, 3, 4, 6, 8, 9, 11 and 16 [14]. Remarkably, the BTV-8 strain present in Europe since 2006 is one of the rare BTV strains capable of inducing clinical signs in cattle due to its pathogenic and transplacental transmission properties [14].

In this study, we investigated BTV and EHDV genomes from 61 EDTA blood samples from domestic goats and cattle from different farms in Oman. These samples were collected from animals with clinical signs of FMD virus in 2020 and 2021. BTV antibodies were tested in 145 sera from sheep (*n* = 23), goats (*n* = 26) and cattle (*n* = 96). Anti-EHDV were also tested in the 96 cattle sera. The results obtained show a circulation of these two viruses in the Sultanate of Oman, with notably the detection of BTV-1, 4, 8, 10 and 16. Partial sequences of segment 2 of BTV-1, 4 and 10 were obtained by Sanger sequencing. A BTV-8 strain was isolated, and the whole genome sequences, determined by NGS, were notably compared with the BTV-8 genomes from strains present in Europe, in the Mediterranean Basin or in Africa and another BTV-8 isolated in Mayotte in 2016. 

## 2. Materials and Methods

### 2.1. Samples

Sera (*n* = 145) and whole blood (*n* = 61) from goats, sheep and cattle were collected from domestic ruminant herds showing clinical signs of FMD in 2020 and 2021. The whole blood samples taken on EDTA were frozen at −80 °C and sent to the ANSES laboratory with dry ice. The sera were stored and sent at −20 °C. Due to the potential presence of FMD virus in the samples, all analyses performed for this study were performed in an A3 containment laboratory.

### 2.2. Serological Analyses

BTV and EHDV antibodies were tested using two VP7 competition enzyme-linked immunosorbent assays (ELISA) (ID SCREEN Bluetongue or EHDV competition kits, Innovative Diagnostics, France), according to the manufacturer’s instructions. On the basis of the ELISAs cut-off values, the tested samples with an inhibition percentage <35% for bluetongue and <30% for EHD were considered positive.

### 2.3. RT-qPCR Analyses

Total RNA from EDTA blood samples was extracted from 100 µL of blood using a Kingfisher 96 robot and the ID extraction kit (Innovative Diagnostics, France) according to the manufacturer’s instructions. Finally, the RNAs were eluted in 80 µL of water, and 5 µL of the eluted RNA previously heated to 95 °C for 3 min was tested with EHDV and BTV group-specific real time RT-PCRs (rt-qPCR). BTV rt-qPCRs were performed using a commercial kit (ADI-352, Bio-X Diagnostics, Rochefort, Belgium). This pan-rt-qPCR BTV kit allows all 36 BTV serotypes to be detected by amplification of BTV S10. EHDV RNA was detected using a pan-EHDV rt-qPCR targeting the S9 [15]. These two pan rt-qPCRs allow specific detection of BTV or EHDV regardless of serotype.

### 2.4. Virus Isolation

The rt-qPCR EHDV and BTV positive blood samples were inoculated to KC cells (*Culicoides sonorensis* cell line) or to embryonated chicken eggs. Briefly, a confluent monolayer of KC cells was inoculated with EDTA-blood samples diluted 1 to 10 in sterile PBS. In parallel, groups of three embryonated chicken eggs were each inoculated intravenously with 0.1–0.2 mL of this diluted blood. For KC cells, the inoculum was removed 24 h after inoculation, and the cells were incubated 7 days with cell culture medium. The dead embryos or KC culture cells were tested by pan EHDV or BTV rt-qPCR and then used to inoculate BSR as already described [16].

### 2.5. Serotype Determination

End-point RT-PCRs were performed in the presence of RNA extracted from EHDV or BTV pan rt-qPCR positive bloods or cell cultures in order to amplify part of S2 of EHDV and BTV. Gel-based subgroup-specific RT-PCRs were described previously [15,16,17]. Commercial type-specific rt-qPCRs kits as well as rt-qPCRs methods published by Maan [18,19], Lorusso [13] and Viarouge [15] were also used for serotyping. These different PCRs were performed in presence of 5 µL of total RNA extracted and denatured at 95 °C for 3 min. After migration of the amplification products on agarose gel, the amplified S2 portions were then sequenced using Sanger technology (Eurofins, France).

### 2.6. Full Genome Sequencing

Next-generation sequencing was performed on total RNA extracts from infected BSR cells using the QiaAmp Viral RNA mini extraction kit (Qiagen, Hilden, Germany) and the Qiacube extractor. Then, 140 µL of infected culture media was extracted without carrier in a final elution volume of 60 µL. One µL of the eluted RNA was tested by BTV pan rt-qPCRs. After a rRNA depletion with the Low Input Ribominus Kit (Ambion, Hongkong, China), the RNA library was performed with the Ion Total RNA-Seq Kit v2 (Life Technologies, Carlsbad, CA, USA), using manufacturer recommendations, then sequenced with the Ion Torrent Proton Life Technologies, Shanghai, China. The resulting reads were cleaned with the Trimmomatic 0.32 software, then a Bowtie 2 alignment was performed on BTV genome references. The reads were down-sampled to fit a global coverage estimation of 80 x and submitted to the SPAdes 3.1.1 de novo assembler. The de novo contigs were then submitted to BLAST (Basic Local Alignment Search Tool, Bethesda, MD, USA) on a local nt database. For each segment, the best matches were selected for a Bowtie 2 alignment producing clean and robust 5′ and 3′ ends. Finally, the de novo assemblies and the alignment on the references were compared, and the strict identities of the de novo and aligned sequences were assessed [20].

### 2.7. Genome Annotation and Phylogenetic Analyses

Each of the 10 segments of the Omani BTV-8 strain described in this study were annotated using Geneious Prime (version 2022.0.2) and deposited in GenBank (accession number (AN) from S1 to S10: OQ860824 to OQ860833). An online BLAST search was used to compare the ten segments of the Omani BTV-8 strain to reference sequences listed in GenBank sequence databases (NCBI) and also to the 10 segments of a Mayotte Island BTV-8 (Indian Ocean) isolated in 2016 in the laboratory and a South African (SA) BTV-8 isolated in 1937. Among the 1000 first hits, we then selected hit sequences according to genomic diversity and a geographical diversity for alignment and phylogenetic analysis using MEGA X (version 10.2.0) [21]. Alignments were first performed using MUSCLE [22], and then phylogenetic trees were reconstructed using the Maximum Likelihood method and Tamura–Nei model, with a bootstrap of 1000 [23]. The tree is drawn to scale, with branch lengths measured in the number of substitutions per site. All positions containing gaps and missing data were eliminated. Further information is provided in the figure legends.

## 3. Results

### 3.1. Serological Results

The seroprevalence for BT in Oman is high (74%) in cattle and illustrates a significant circulation of BTV in the country (Table 1). It seems less important in small ruminants with 50 and 30.4% of goats and sheep, respectively, showing antibodies against BTV (Table 2). Seroprevalence is also high for EHDV: more than one out of two cattle has antibodies against this virus (Table 1). In total, 46 (90.2%) of the 51 cattle with antibodies to EHDV were also positive in BTV ELISA.

### 3.2. Pan rt-qPCR Results and Virus Isolation

In total, 15 of the 61 animals tested were positive for BTV rt-qPCR (Ct value range: 22 to 34), and 11 of the 56 cattle tested were positive for pan EHDV rt-qPCR (Ct value range: 30 to 35). Three cattle were positive for both viruses. Viral isolation assays on embryonated eggs and KC cells were tested several times on each rt-qPCR-positive blood sample. Only one isolate of BTV was obtained, and no EHDV isolate was recovered.

### 3.3. Typing and Sequencing

The Figure 1 illustrates the location of the herds where BTV and EHDV genome detection occurred. The rt-qPCRs from Maan et al. [18,19], performed on the RNA extracts from the blood, have shown the presence of the BTV-1, 4, 8, 10 and 16. The end-point RT-PCRs [16] performed on the RNAs from the bloods allowed the determination of partial sequences of S2 of BTV-1, 4 and 10. No S2 sequence could be obtained for BTV-16-positive blood by classical RT-PCRs. The complete genome sequences of the BTV-8 strain were obtained by NGS.The EHDV type specific rt-qPCRs as well as the EHDV end-point RT-PCRs did not yield positive result.

### 3.4. BTV Genome Annotation and Phylogenetic Analyses

S2 partial sequences were obtained for BTV-1, 10 and 4 from pan-BTV rt-qPCR-positive samples tested with end-point RT-PCRs. These sequences were compared with homologous segments in GenBank (Table 3).

The complete length of the S1, 2, 3, 4, 6, 7, 8, 9 and 10 were obtained for the BTV-8 genome, with the usual 5′ and 3′ extremities sequences (5′- GTTAAA … ACTTAC-3′). Regarding S5, only three nucleotides were missing at the 3′ extremity. Thus, all expected CDS were recovered (Table 4).Table 4 shows the sequence homologies obtained between the 10 segments of the BTV-8 Oman strain and 3 other BTV-8 strains: one isolated in France in 2015 (AN: MN495893 to MN495902), the second isolated in Mayotte (Indian Ocean) in 2016 (AN: OQ860834 to OQ860843) and a 3rd isolated in SA in 1937 (AN: MT078269 to MT078278). Table 4 also shows the best homology found between BTV-8 Oman sequences and homologous sequences deposited on GenBank. Sequence alignments (in AA) of the VP3, NS1, VP5, VP7 and NS3 of the four BTV-8 strains (Table 4) show more than 99% for all these VPs.

**Table 4 viruses-15-01259-t004:** Annotation of the BTV-8 OMAN RNA segments/VP and homology comparison with other BTV segments deposited in GenBank.

					% nt and (AA) Homology between BTV8 Oman and			
Seg.	AN	Length bp	CDS (nt Position/Length AA)	Product	BTV-8 France	BTV-8 Mayotte	BTV-8 SA	(%)—Better Homology in GenBank (AN)
1	OQ860824	3944	12–3920/1302	VP1	93.6 (98.8)	92.5 (98.6)	96 (99.1)	97.8—BTV-1 Isr 2019 (OM502362)
2	OQ860825	2939	18–2903/961	VP2	94.6 (97.2)	98.7 (99)	92.5 (97.1)	95.9—BTV-8 Nig 1982 (AJ585184)
3	OQ860826	2772	18–2723/901	VP3	94.5 (99.7)	94.3 (99.7)	95.5 (99.8)	97.1—BTV-24 SA 2017 (MG255591)
4	OQ860827	1982	9–1943/644	VP4	94.2 (97.8)	94 (98.4)	95.2 (98.8)	97.5—BTV-3 Isr 2016 (MG344993)
5	OQ860828	1773	35–1693/552	NS1	93.7 (99.3)	96.9 (99.3)	93 (99.1)	97.7—BTV-4 Fr 2020 (OK018214)
6	OQ860829	1637	28–1608/526	VP5	96.8 (99.8)	96.6 (99.8)	96.2 (99.6)	96.9—BTV-8 Nig 1982 (AJ586706)
7	OQ860830	1156	18–1067/349	VP7	97.3 (100)	94.7 (99.4)	95.1 (100)	97.7—BTV-1 Sud 1987 (KP821626)
8	OQ860831	1125	20–1084/354	NS2	92.9 (96.9)	92.4 (97.7)	94.7 (97.5)	97.5—BTV-1 Isr 2019 (OM502359)
9	OQ860832	1049	16–1005/329	VP6	95.9 (94.8)	95 (93.0)	95.3 (95.5)	96.8—BTV-24 SA 2020 (MT090653)
			182–415/77	NS4	98.7 (98.7)	98.7 (100)	98.7 (100)	
10	OQ860833	822	20–709/229	NS3	95.7 (100)	98.6 (99.6)	92.3 (99.1)	98.8—BTV-15 Isr 2006 (JX272377)
			108–287/59	NS5	93.9 (100)	93.3 (99.6)	93.3 (99.1)	

The phylogenetic tree of S2 (Figure 2) was performed with selected sequences of BTV-8, 18 and 23 that constitute the nucleotype D of the BTV S2 [24]. The tree shows two clusters of S2 of BTV-8: the Oman and Mayotte S2, which are very similar, form one of these clusters also consisting of S2s of European, Mediterranean and Nigerian BTV-8 strains. The second cluster includes the S2 of the SA and Kenia strains.

**Figure 2 viruses-15-01259-f002:**
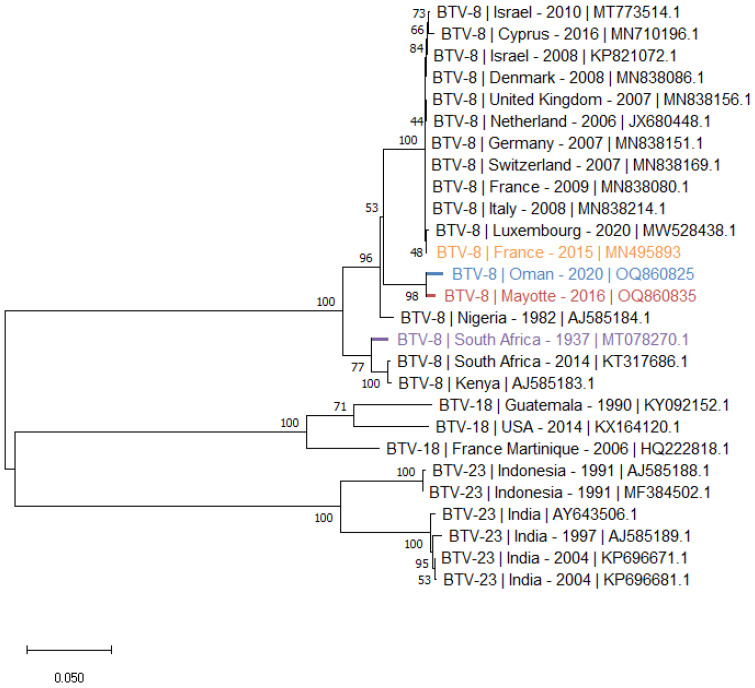
**Phylogenetic analysis of BTV S2 sequences**. Phylogenetic analysis of S2 sequences of BTV-8, 18 and 23 strains using the maximum likelihood method and Tamura–Nei model with 1000 bootstrap replicates in MEGA X. This analysis involved 27 nucleotide sequences, and there were a total of 2943 positions in the final dataset. Bootstrap values appeared at the corresponding nodes. In the phylogenetic tree, GenBank sequences, bluetongue serotype, country and year of sample collection are given. The sequences investigated in the present study are marked in blue, red, orange and purple.

The S6 phylogenetic tree (Figure 3) illustrates the homologies between S6 of BTV-8 and BTV-18, constituting the G nucleotype of BTV VP5 [25]. As observed with the S2 phylogenetic tree, the S6 of the Oman and Mayotte strains are closely related and form a cluster with BTV strains S6 isolated in Europe, in the Mediterranean basin and in Nigeria. In the cluster of the BTV-8 S6, two BTV-18 S6 from SA were found having a strong homology with BTV-8 S6 from the same area. These two SA BTV-18 S6 are clearly distinct from those isolated from American strains (Figure 3).

**Figure 3 viruses-15-01259-f003:**
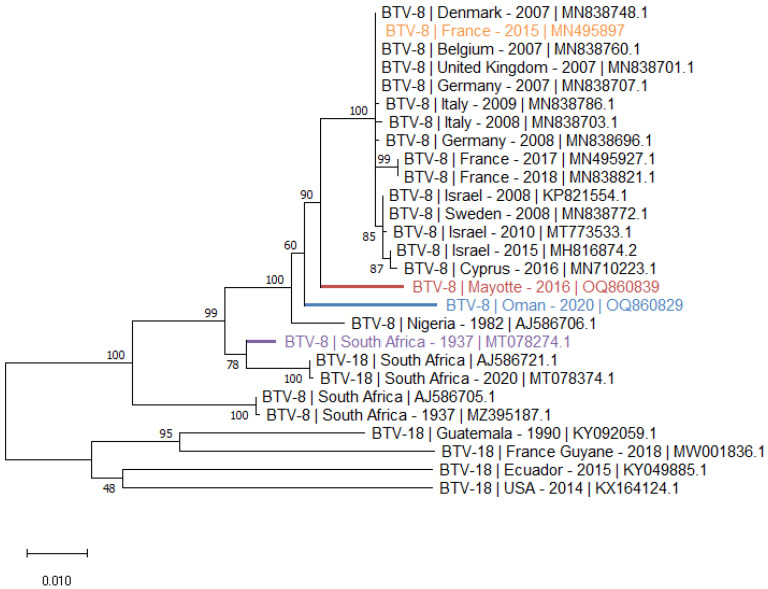
**Phylogenetic analysis of BTV S6 sequences**. Phylogenetic analysis of S6 sequences of BTV-8 and 18 strains using the Maximum Likelihood method and Tamura–Nei model (1000 replicates). This analysis involved 27 nucleotide sequences, and there were a total of 1591 positions in the final dataset. In the phylogenetic tree, GenBank sequences, species designations and strain names are given. The sequences investigated in the present study are marked in blue, red, orange and purple.

S1, 3 to 5 and 7 to 10 of the Oman strain have the strongest homologies with segments all originating from BTV of many serotypes strains circulating further west: in the Mediterranean basin (Israel, Lebanon, Cyprus, Maghreb...), in the European countries, in Central Africa (Sudan, Nigeria, Cameroon) or SA (See Appendix A for phylogenetic analysis of each BTV segment). Note the presence of a S4 of a Chinese BTV-7 strain in the cluster otherwise presenting only BTV S4 from strains isolated in Africa, Europe or the Arabian Peninsula (Appendix A), suggesting for this Chinese BTV-7 segment 4 a “western” origin.

## 4. Discussion

In this study, serological results on bovine samples show that 74% of the animals are seropositive against BTV and 53.1% against EHDV, suggesting a significant circulation of these two viruses in the south-eastern Arabian Peninsula. In comparison, BTV ELISA results obtained on small ruminants seem to show a lower seroprevalence (50% and 30.4% of positive results for goat and sheep, respectively). This result, to be taken with caution because few small ruminants were tested in this study, seems, however, to illustrate that cattle are better sentinels for BTV and EHDV circulation. BTV surveillance can be focused on cattle as their larger body size results in a greater range of attraction to Culicoides than small ruminants and are therefore more likely to be involved in virus transmission [26,27]. The low seroprevalence rates of EHDV reported in small ruminants in previous studies, compared to those observed in cattle, explain why small ruminant samples were not tested in our study for either EHDV antibodies or genomes [28].

Despite our efforts, the EHDV serotype(s) present in the Sultanate of Oman samples remained undetermined. EHDV typing assays based on specific rt-qPCRs were performed on pan-EHDV-positive samples, and none of them gave a positive result. The circulation of an EHDV-8 could be ruled out as the rt-qPCR used in this study was developed to specifically detect the EHDV-8 strain recently described in the Maghreb and in southern Europe [12,13]. Concerning the other rt-qPCRs used for EHDV typing assays, these tools were developed 10 years ago from EHDV S2 sequences deposited in GenBank at this time. Thus, primers and probes used in these assays might not be able to detect divergent EHDV S2 sequences. Finally, end-point PCR, allowing amplification of S2 portions of EHDV belonging to the same nucleotype, also failed [11]. Here, again, the lack of sensitivity if sequence mismatches are present, as well as the low amount of EHDV viral genome detected in the blood (Ct values in pan-rt-qPCR between 30 and 34), could explain the absence of amplification.

In total, 15 of 61 animals tested were pan-BTV rt-qPCR-positive. Serotype 4 is the most detected serotype, with 7 cattle detected positive (out of 15) in BTV-4 specific rt-qPCR, located in different farms (Figure 1). Two animals from the same farm located in northern Oman were positive for BTV-8. Despite repeated viral isolation tests on these 15 BTV-positive samples, only one BTV-8 strain could be obtained and its genome fully sequenced. The presence of BTV-1, 4, 8, 10 and 16 was determined using the rt-qPCRs developed by Maan et al. [18], demonstrating the robustness of this method and its efficiency. Partial S2 sequences of BTV-1, 4 and 10 were obtained in classical RT-PCRs from the blood samples and sequenced by the Sanger method.

The genomic segments of the Mayotte and Oman BTV-8 strains have their strongest homologies with the multiple BTV serotypes that have circulated or are still circulating in Africa, in the Mediterranean basin and in Europe (Table 3 and Table 4, Figure 2 and Figure 3, and the Appendix A). None of the 10 segments of the BTV-8 Oman strain appear to have an eastern, Indian origin. This is also the case with the amplified partial S2 of BTV-1, 4 and 10. The highest homologies of homologous sequences from GenBank with the amplified BTV-1 and 4 partial S2 are with BTV-1 and 4 strains that circulated in the Mediterranean basin (Table 2). The BTV-10 S2 circulating in Oman has the highest nt homology with the SA vaccine strain (93.6% nt).

The S6 phylogenic tree shows two BTV-18 S6 having 98% of nt homology with the S6 of SA BTV-8. The VP5 of these BTV-18 have a homology of more than 99% with VP5 of the different BTV-8 strains, showing a common origin for these S6. The conformation of the VP2s of these serotypes 8 and 18, belonging to the same nucleotype, allows these VP2s to be associated with the same VP5.

Many BTV-8 strains circulating since 2006 in Europe and the Mediterranean basin, including Cyprus and Israel, have been isolated, sequenced and had their genomes deposited in GenBank. However, all of these BTV-8 strains have a common origin whose first isolate was detected in the Netherlands in 2006 [29]. They are very similar, and the strains circulating in Europe today are identical to the 2006 prototype. In Europe, few serotypes co-circulate at the same time and place. However, in 2008, it has been described that a wild BTV-8 strain isolated in France had reassorted with genomic segments of BTV-1 circulating in the same region [30], but this reassorted strain does not appear to have circulated in the field. The known reassortant BTV-8 strains that have circulated have been observed in Israel, where multiple BTV serotypes co-circulate, increasing the likelihood of reassortant strains [31]. The other genomic sequences of BTV-8 strains found in GenBank are African strains, mostly from SA, and they have between them an important genomic homology showing a common origin as illustrated by the phylogenic trees of S2 and 6. For these reasons, in our study, the genomic sequences obtained from the Oman and the Mayotte BTV-8 strains were compared with sequences of one BTV-8 prototype that circulated in Europe/Mediterranean Basin (i.e., BTV-8 France isolated in 2015) and one in SA isolated in 1937 (Table 3 and Table 4).

The sequences of the BTV-8 strains from Oman and Mayotte add to the data on this serotype. This study shows that this serotype also circulates in the Indian Ocean, that their genomes are close and suggest that all these BTV-8 strains (from the Middle East region, Mayotte, Africa, Europe and the Mediterranean basin) have a common African origin.

## Figures and Tables

**Figure 1 viruses-15-01259-f001:**
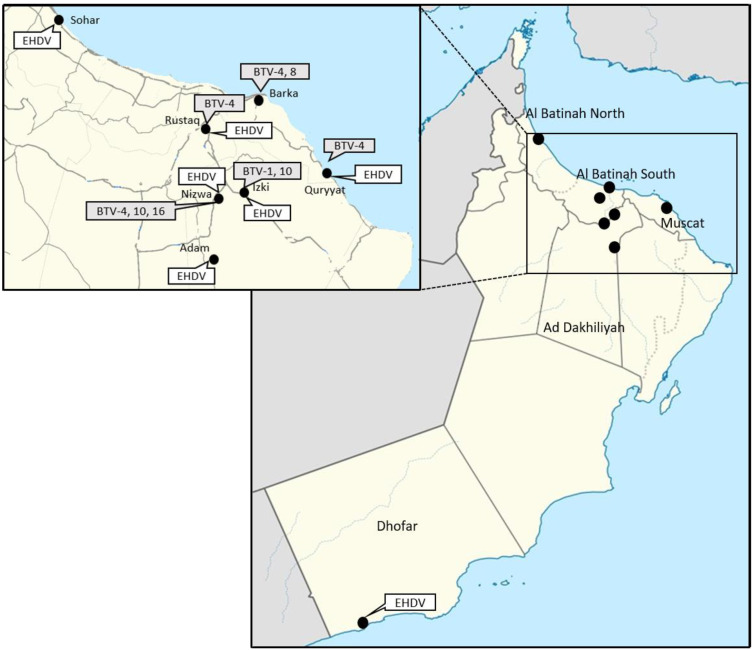
Location of the BTV serotypes and EHDV positive samples.

**Table 1 viruses-15-01259-t001:** EHD and BTV ELISA results in cattle.

	BTV ELISA			
	Negative	Doubtful	Positive	Number Tested
number	23	2	71	96
%	24.0	2.1	74.0	
	**EHDV ELISA**			
number	44	1	51	96
%	45.8	1	53.1	

**Table 2 viruses-15-01259-t002:** BTV ELISA results in goats and sheep.

		BTV ELISA			
		Negative	Doubtful	Positive	Number tested
Goats	number	13	0	13	26
	%	50	0	50	
Sheep	number	15	1	7	23
	%	65.2	4.3	30.4	

**Table 3 viruses-15-01259-t003:** Annotation of the BTV-1, 4 and 10 OMAN S2 and homology comparison with other BTV S2 deposited in GenBank.

AN	Length bp	CDS (nt Position/Length AA)	Product	(% nt/AA)—Better Homology in GenBank (AN)
OQ860822	342	20–335/114	Partial VP2 BTV-1	(98.0/98.2)—BTV-1 Spa 2008 (KP821020)
OQ860823	318	1182–1523/105	Partial VP2 BTV-10	(93.4/99.0)—BTV-10 SA 2017 (MT078290)
OQ860904	2874	20–2890/956	Partial VP2 BTV-4	(97.5/98.6)—BTV-4 Spa 2003 (KP821067)

## Data Availability

The data presented in this study are available on request from the corresponding author.

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
