# Peer review of "Circulation of Bluetongue Virus Serotypes 1, 4, 8, 10 and 16 and Epizootic Hemorrhagic Disease Virus in the Sultanate of Oman in 2020–2021"

_viruses, 2023, doi:10.3390/v15061259_

Round 1

Reviewer 1 Report

In the manuscript entitled " Circulation of bluetongue virus serotypes 1, 4, 8, 10 and 16 and epizootic haemorrhagic disease virus in the Sultanate of Oman in 2020 - 2021" by Bréard et al., the authors present the recent circulation of several serotypes of BTV and EHDV in the Sultanate of Oman. As Middle East undergoes frequent outbreaks of different serotypes and could be an entry route of BTV and EHDV to Europe, updated circulation data are of significant value.

  The protocol, although classical is described and detailed enough. Results are clear and support the conclusions.

General comment: Please be consistent with the way you refer to “BTV8”, as it is sometimes written “BTV-8” throughout the manuscript.

Specific comments :

Title : should not be written with a final point.

Authors list: seems not to be properly aligned on the left, please verify.

L.97: “This kit allows all 36 BTV serotypes to be detected”, please rephrase to stress the unspecificity of the kit and its “pan-BTV” detection

L.124 & 131:  BLAST is mentioned L.124 but the acronym is detailed L. 131, please provide details for BLAST at first use.

L.146: “the BTV” è BTV

L.152: the authors do not mention either in the Table 2 or in the text the results of EHDV ELISA on small ruminants. The M&M section suggests both large and small ruminants were tested though. Please provide this information.

L.220: BT8 è BTV8

L.233: I would advise to change “indicating” to “suggesting”, as BTV and EHDV seroprevalence can be quite long lasting hence could only be the reflection of a past significant circulation. This is backed up by the high Ct values reported for EHDV L. 255.

L. 272: “S2circulatingè S2 circulating

The paper is very readable and I have only a few comments regarding style and language.

Author Response

General comment: Please be consistent with the way you refer to “BTV8”, as it is sometimes written “BTV-8” throughout the manuscript.

Specific comments :

Throughout the text, it is now written BTV- to be consistent with the phylogenetic trees.

Title: should not be written with a final point.

It has been deleted.

Authors list: seems not to be properly aligned on the left, please verify.

We have verified this point.

L.97: “This kit allows all 36 BTV serotypes to be detected”, please rephrase to stress the unspecificity of the kit and its “pan-BTV” detection.

We have clarified this point lines 98 – 100. “This pan- rt-qPCR BTV kit allows all 36 BTV serotypes to be detected by amplification of BTV S10. EHDV RNA was detected using a pan-EHDV rt-qPCR targeting the S9 [15]. These two pan rt-qPCRs allow specific detection of BTV or EHDV regardless of serotype.”

L.124 & 131: BLAST is mentioned L.124 but the acronym is detailed L. 131, please provide details for BLAST at first use.

It has been modified.

L.146: “the BTV” è BTV

It has been modified.

L.152: the authors do not mention either in the Table 2 or in the text the results of EHDV ELISA on small ruminants. The M&M section suggests both large and small ruminants were tested though. Please provide this information.

Details have been added on lines 70-74 regarding animals tested by EHDV ELISA and, on lines 253-255, we explain why small ruminants were not tested for EHDV genome or antibody.

L.220: BT8 è BTV8

It has been modified.

L.233: I would advise to change “indicating” to “suggesting”, as BTV and EHDV seroprevalence can be quite long lasting hence could only be the reflection of a past significant circulation. This is backed up by the high Ct values reported for EHDV L. 255.

It has been changed.

  1. 272: “S2circulating” è S2 circulating

It has been modified.

Reviewer 2 Report

The manuscript entitle ¨Circulation of bluetongue virus serotypes 1, 4, 8, 10 and 16 and  epizootic haemorrhagic disease virus in the Sultanate of Oman in 2020 – 2021¨ investigates BTV and EHDV genomes from 61 EDTA blood samples from domestic goats and cattle from different farms in Oman. They also analyze anti-EHDV and BTV antibodies in 145 sera from sheep, goats and cattle. The results showed the circulation of these two viruses in Oman, and the detection of BTV-1, 4, 8, 10 and 16. In addition, the authors isolated a BTV-8 strain and they compare the whole genome sequences, determined by NGS, with the BTV-8 genomes from strains present in Europe, in the Mediterranean Basin or in Africa, and another BTV-8 isolated in Mayotte in 2016. The study is interesting, the methodology is adequate and the conclusions are consistent with the experimental results obtained.

This epidemiological study will be of great interest in the field of orbiviruses in particular and in the field of arboviruses in general.

Minor editing of English language required.

Author Response

As no specific requests were made by reviewer 2, only changes in the text resulting from the editor's or reviewer 1's remarks were made.